# Bamboo Construction Inspired by Vernacular Techniques for Reducing Carbon Footprint: A Life Cycle Assessment (LCA)

**Carlos Eduardo Rincón** [1], **Jorge Augusto Montoya** [1] **and Hector F. Archila** [2,*]

1   Facultad de Ciencias Ambientales, Universidad Tecnológica de Pereira, Pereira 660017, Colombia; carlos.rincon@utp.edu.co (C.E.R.); jorgemontoya@utp.edu.co (J.A.M.)
2   School of Architecture and Environment, College of Arts, Technology and Environment (CATE), University of the West of England, Bristol BS16 1QY, UK
*   Correspondence: hector.archila@uwe.ac.uk

**Abstract:** Whilst upcoming innovations on digital technology and renewable energy can have a significant impact on the reduction of operational carbon emissions in the construction industry, readily available fast-growing building materials like bamboo are already proving reductions in the embodied carbon of dwellings above 60% when compared to traditional brickwork in Colombia. This paper presents a like-by-like comparison of the environmental impact of a conventional clay brick house (CBH) and a bamboo house for social housing in Colombia, which was built using adapted vernacular technologies. The bamboo house uses bamboo species *Guadua angustifolia* Kunth as the main structural support for the light cement bamboo frame (LCBF) system, a.k.a. 'cemented bahareque', whilst the CBH combines clay bricks and steel for the load-bearing walls. Traditionally built *Guadua angustifolia* Kunth bahareque (GaKB) houses are a key part of the vernacular architecture in the 'coffee cultural landscape of Colombia' (CCLC) recognised by UNESCO. A life cycle assessment (LCA) was performed to calculate the carbon footprint of the houses following four phases: (1) definition of objective and scope; (2) inventory analysis; (3) impact assessment; and (4) interpretation of results. The results show that the carbon footprint of the GaKB house accounts for about 40% of the CBH, i.e., the GaKB generates a carbon footprint of 107.17 $CO_2$-eq/m$^2$ whilst the CBH results in a carbon footprint of 298.44 kg $CO_2$-eq/m$^2$. Furthermore, from a carbon balance calculation, the carbon footprint of the GaKB house is further reduced to about 36% of the CSB house. LCA results for the built GaKB house demonstrate that vernacular housing projects that preserve cultural heritage can also be resilient and climate-neutral. This paper sets a precedent for the establishment of targeted government policies and industry practices that preserve the cultural heritage and vernacular technologies in the CCLC region and in other emergent economies worldwide whilst promoting future-proof and net-zero carbon construction.

**Keywords:** biobased building materials; life cycle assessment; carbon footprint; environmental loads; sustainable building; vernacular housing



## 1. Introduction

Construction with bamboo and bio-based materials is a key part of the cultural heritage of the 'coffee cultural landscape of Colombia' (CCLC), which was inscribed on the UNESCO world heritage list in June 2011. The CCLC region includes 18 urban centres and six farming areas (highlighted in Figure 1) across four neighbouring states (departments) in Colombia (Caldas, Risaralda, Quindío, and Valle del Cauca). However, this heritage is at risk of disappearing due to government policies promoting the use of conventional "modern" materials such as structural clay brick, cement block, steel, and concrete, which are known to contribute to about 16% of global carbon dioxide ($CO_2$) emissions. With the global demand for steel and cement to grow between 12–23% and 15–40%, respectively, by 2050 [1], the need to rescue and future-proof low-carbon vernacular techniques widespread in the CCLC region and worldwide is paramount.

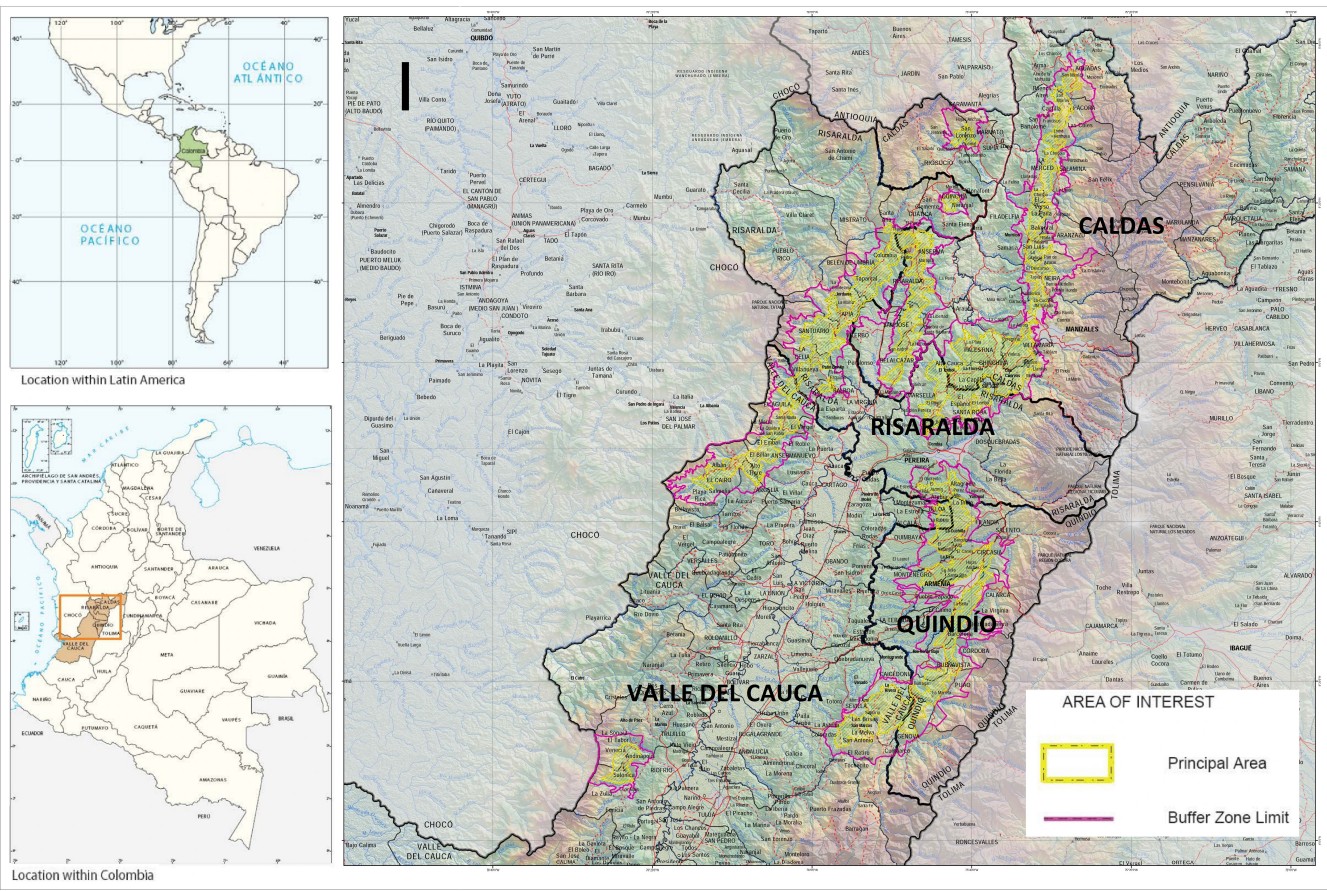

**Figure 1.** Location of the Colombian Coffee Cultural Landscape (CCLC), which includes the departments (states) of Caldas, Risaralda, Quindío, and some municipalities of northern Valle del Cauca.

Recent research on 'sustainable vernacular architecture' is demonstrating how the implementation of vernacular techniques and the use of natural and bio-based materials globally offer vital lessons for sustainable architecture, with reduced environmental impacts and increased energy efficiency [2–4].

A vernacular technique in the CCLC region commonly known as 'bahareque' [5,6] makes up a high degree of vernacular dwellings in the region, which is one of the 16 attributes that give the CCLC its exceptional character, according to UNESCO. Traditionally, the bahareque system is composed of bamboo mats ('*esterilla*') and/or metal mesh (chicken mesh) nailed to a bamboo frame, which is then rendered with a lime or cement mortar render, as seen in Figure 2. A modern adaptation of the system in use in Colombia, the Philippines, and other parts of the world is referred to as the light cement bamboo frame (LCBF) system and recognised as a structural shear wall system for low-rise construction by ISO 22156:2021 [7–10].

Nevertheless, current perceptions of bahareque construction are torn between two positions: on the one hand, the stigmatisation due to its association with informal housing processes in the urban peripheries (slums) and, on the other hand, its recognition as both a standardised 'seismic-resistant' building system (LCBF) under ISO 22156:2021 and a sustainable construction system for two- to three-storey dwellings [6,9,11–13].

After a devastating earthquake with epicentre in the CCLC region in 1999, the modern version of bahareque, the LCBF or cemented bahareque wall system was included in the Colombian code of earthquake-resistant construction or NSR-10 [14,15]. Recently, the use of cemented bahareque has been included in the Andean Standard N-2015 [16], ISO 22156:2021 [9], and other building codes around the world, but despite this, the use of LCBF in Colombia has not been sufficiently stimulated by government agencies. On

a sight of hope, a concern for rescuing vernacular techniques and ancestral knowledge inherent to architectural heritage in Colombia is becoming evident in public policies such as the 'Housing and habitat Law' (HhL) and the National Sustainable Buildings Policy (NSBP). In particular, the HhL goes beyond the concept of social interest housing (affordable housing) by introducing the concept of cultural interest housing (CIH), which is described as "being totally rooted in its territory and its climate; with a design, construction, financing and regulatory criteria responding to local customs, as well as following local traditions, lifestyles, materials, construction and production techniques" [17]. On the other hand, the NSBP "seeks to promote the inclusion of sustainability criteria within the life cycle of buildings, through instruments for transition, monitoring and control, and financial incentives that allow the implementation of sustainable construction initiatives" [18]. Whilst implementation of these policies is lacking, recent research on sustainable construction in Colombia includes studies on circular economy [19], lean construction to optimise process and efficiency [20], and LCA of traditional and engineered biobased bamboo buildings [9]. Circular economy and LCA approaches applied to vernacular building systems can take into account end-of-life scenarios for the constituent materials, encourage the efficient use of biobased materials, and promote the recovery of degraded land and ecological systems.

1 Bamboo studs (round GaK culms)

2 Bamboo top rail (round GaK culms)

3 Bamboo rafters (round GaK culms)

4 *Esterilla* (GaK mats)

5 Cement mortar render

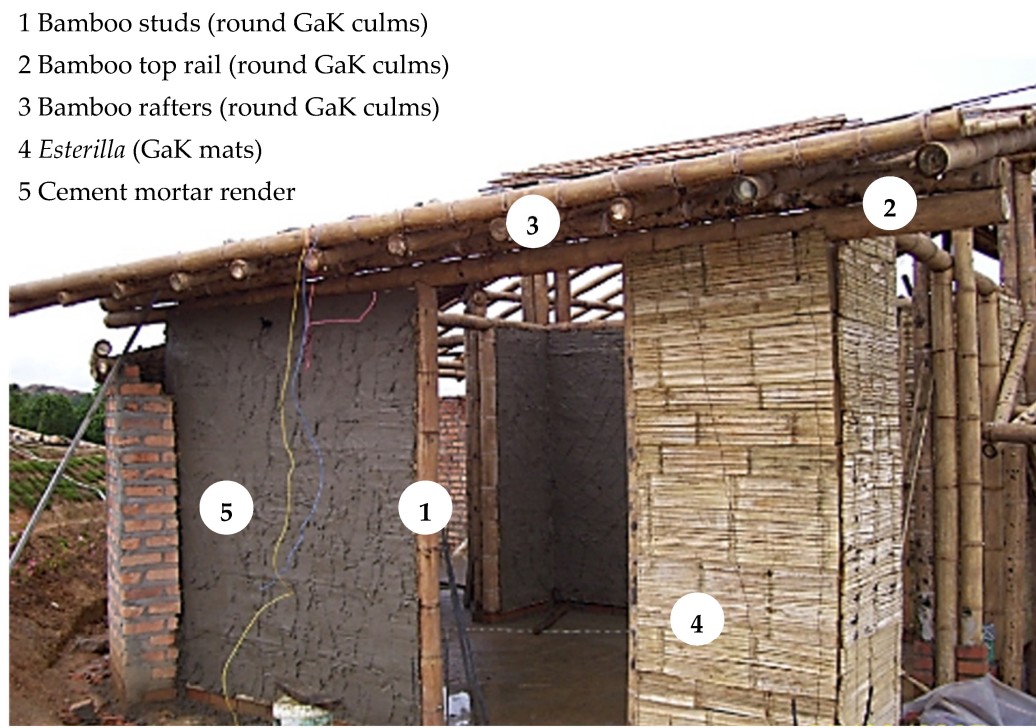

**Figure 2.** Light cement bamboo frame (LCBF) system.

The life cycle assessment (LCA) approach is recognised worldwide for its effectiveness in assessing environmental impacts. In particular, the measurement of the equivalent carbon footprint (kg $CO_2$-eq) of materials and buildings serves as a universal environmental indicator for comparing construction practices [7,8,21,22]. A clear estimation of the impact of these building systems can aid in the establishment of targeted government policies that preserve cultural heritage whilst reducing the carbon emissions of new housing projects in the CCLC region and beyond.

Commonly, the scenarios for LCA studies are fictitious and based on assumptions about materials' utilisation, construction methods, transport efficiencies, etc. A key novelty of this paper is its use of first-hand data from the construction of the vernacular housing model for a comparative LCA. Between 2020 and 2022, a prototype rural house using the modern bahareque system was designed and built on the campus of Universidad Tecnológica de Pereira (UTP) in Pereira, Colombia [23] and closely monitored by one of

the authors, which provided an outstanding test base for this study. This project led by Fedeguadua (the Colombian Guadua bamboo trade association or 'Federación Nacional de Empresarios de Guadua y Bambú') aims to enhance the value of regional bahareque architecture and promote the construction of sustainable, resilient, and affordable vernacular housing for the CCLC. With the aim of aiding the promotion and implementation of heritage and sustainable construction policies in Colombia and beyond, this study provides a systematic analysis of the carbon footprint of a prototype vernacular bahareque house and compares it with that of a widespread conventional clay-brick house in Colombia. Today, the built bahareque house is a space for knowledge transfer and dissemination in the region.

## 2. Materials and Methods

The methodology adopted for calculating the carbon footprint of the selected houses is framed in the LCA approach regulated by ISO 14040:2006 [24] and ISO 14044:2006 [25] standards. The methodological development of this study was carried out in four phases: (1) definition of objective and scope; (2) process inventory analysis; (3) impact assessment; and (4) interpretation (Figure 3). The LCA was carried out with the support of the software openLCA v 1.10.3 and the ecoinvent 3.6 database. The LCA was undertaken using the Intergovernmental Panel on Climate Change (IPCC) 2013 GWP 100th V1.03 impact assessment method [26].

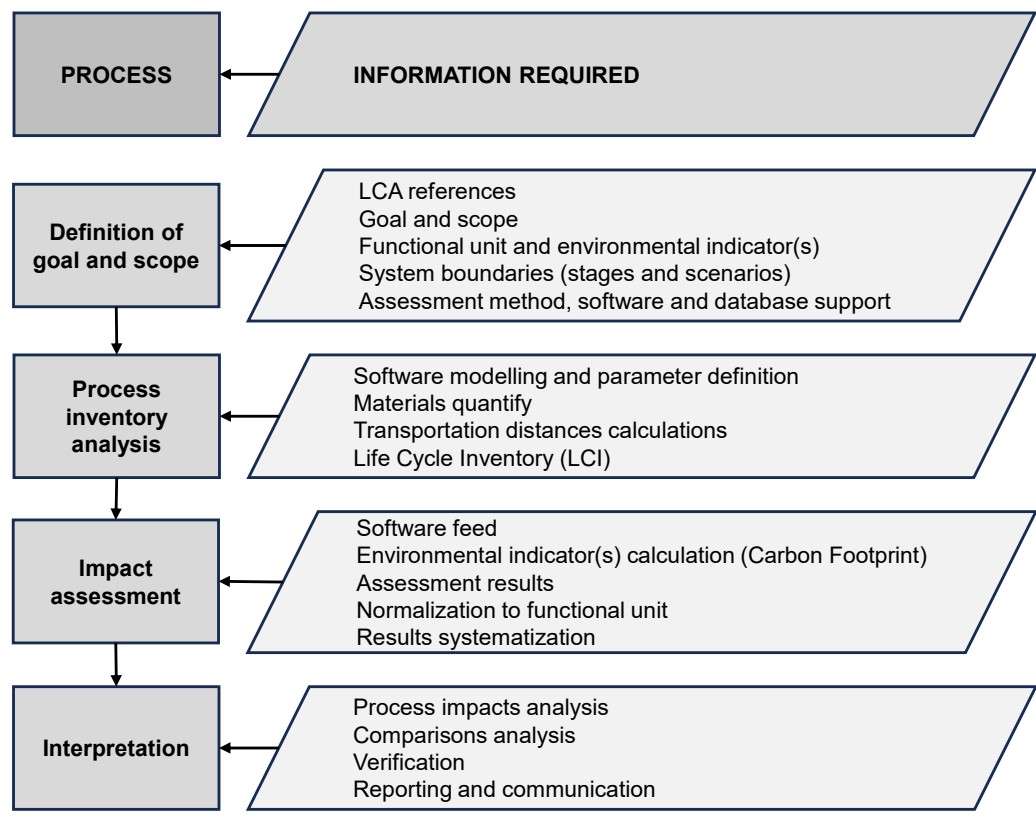

**Figure 3.** The study's methodology flowchart.

### 2.1. Definition of the Goal and Scope of the Life Cycle Assessment (LCA)

#### 2.1.1. Goal and Scope

The goal of the LCA was to quantify the environmental impact in terms of the carbon footprint calculation of two houses with different construction systems for comparison: a conventional social house of Clay Structural Brick (CSB) and a *Guadua angustifolia* Kunth *Bahareque* (GaKB) house built on the UTP campus (Figure 4). The scope of the LCA focused on the supporting elements of the houses, i.e., key components within the construction

system such as the foundation, the floor slab, the structural walls, and the roof structure. Both houses use reinforced concrete strip foundation and reinforced concrete ground floor slabs; CSB uses concrete infilled clay structural walls, reinforced concrete tie beams and crown beams, and a roof structure with steel profiles, while GaKB house uses structural walls on a LCBF system and a roof structure composed of Guadua trusses and beams.

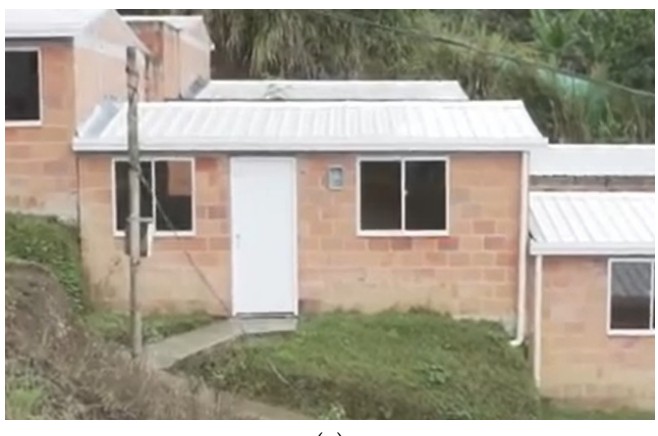 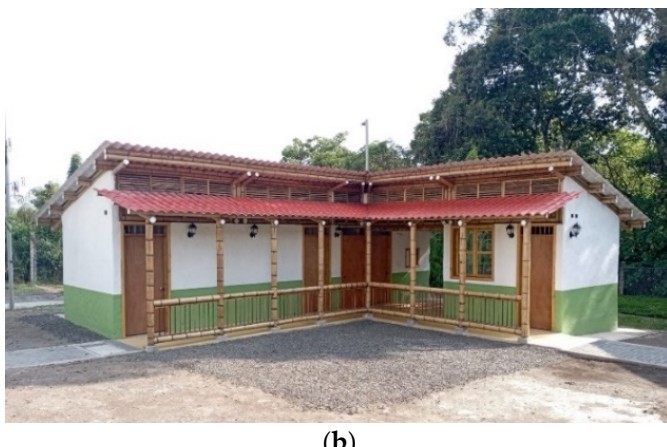

(**a**) (**b**)

**Figure 4.** Photos of the houses under comparison: (**a**) CSB house (EDUR); (**b**) GaKB house at the UTP campus.

2.1.2. Functional Unit and System Boundaries

To establish the functional unit, references were adopted from previous research on LCA, in which alternative housing systems, especially bamboo, were analysed [8,22,27]. The functional unit was determined as kilograms of carbon dioxide equivalent per square metre (kg $CO_2$-eq/m$^2$) of living area.

Regarding system boundaries, two stages were considered according to the European Standard EN 15978:2011 [28], (1) A1–A3 Production stage and (2) A4–A5 Construction process stage, which include five phases: (A1) Supply of raw materials, (A2) Transport, (A3) Manufacturing, (A4) Transport, and (A5) Construction–installation process, where A4 and A5 are modelled scenarios, as seen in Figure 5. The limits of the system did not include the use stage (B1–B7), nor did they include the final stage of the life cycle (C1–C4). This is due to the uncertainty inherent in housing in Latin America due to regulatory gaps regarding the usage and final disposal of construction materials and economic aspects that affect the extension of the life cycle [8].

The functional unit (kg $CO_2$-eq/m$^2$) was chosen to represent the characteristics of each building system. To contrast the carbon footprint, a conventional CSB social housing typology was taken as a baseline; considering the high replicability of this typology in the region and in the country, we worked with the designs of the Bello Horizonte project located in the department of Risaralda, which has 41.04 m$^2$, provided by Empresa de Desarrollo Territorial Urbano y Rural de Risaralda or EDUR (Urban and Rural Development Company of Risaralda) (Figure 6a). For the analysis of the GaKB house, the vernacular housing prototype developed by Fedeguadua was utilised, which has an area of 60.65 m$^2$ (Figure 6b).

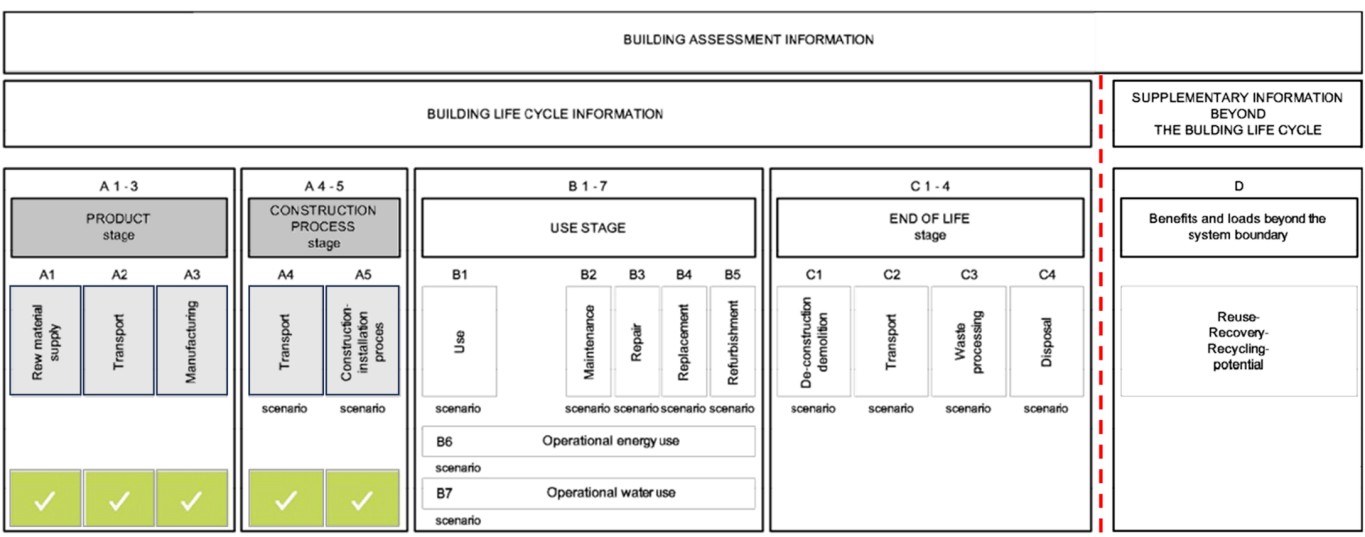

**Figure 5.** Life cycle phases according to EN 15978:2011.

Regarding the measurement of the carbon footprint, the IPCC 2013 GWP 100th V1.03 [26] impact assessment method of the Intergovernmental Panel on Climate Change IPCC 2013 GWP 100th V1.03 was chosen, which allows the analysis of the global warming category.

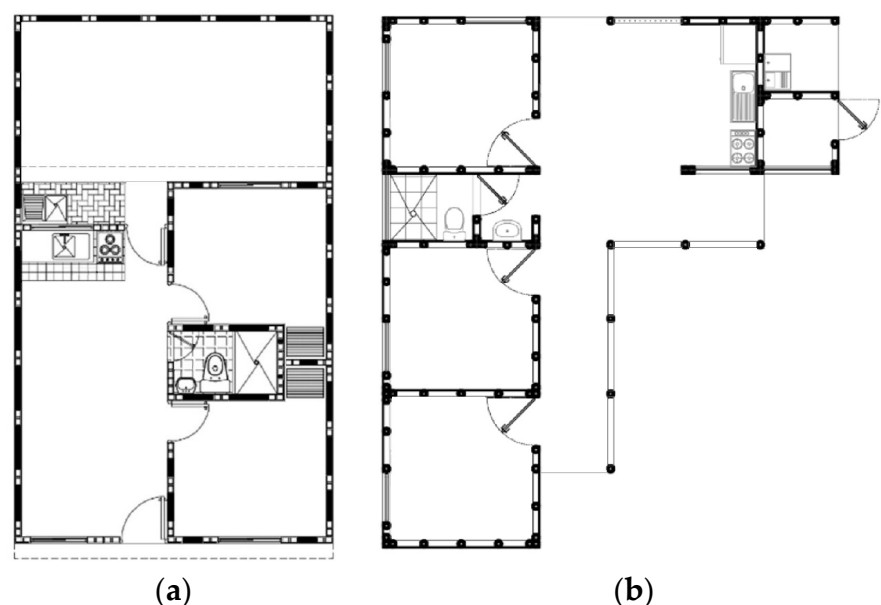

(**a**)                                                  (**b**)

**Figure 6.** Floor plans of the houses: (**a**) CSB house A = 41.04 m$^2$ (EDUR); (**b**) GaKB house A = 60.65 m$^2$.

### 2.2. Process Inventory Analysis

Once the openLCA model was set up and the main parameters were defined, the next step of inventory analysis of the two dwellings was the calculation of material quantities. For this purpose, the architectural and structural designs and the general construction programme and budget were used. The values corresponding to the processes of extraction, transport, and transformation of *Guadua angustifolia* Kunth (GaK) components were based on previous LCA studies [29,30].

### 2.2.1. Transportation Distance Calculations

The LCA model considers the quantities of construction materials and production sites, which were geographically located in Colombia (Table 1); distances were calculated

with the help of the Google Maps route calculation engine. To make the models comparable, the CSB house under analysis was assumed to also be located at the UTP campus.

**Table 1.** Production sites and material transport distances.

| Inputs—Materials | Place of Origin or Production | Distance to UTP Campus (km) |
|---|---|---|
| Reinforcing steel | Acerías Paz del Rio, Duitama, Boyacá | 530 |
| Cement | Yumbo, Valle del Cauca | 203 |
| Washed sand | Corregimiento Caimalito, Pereira, Risaralda | 58.2 |
| Gravel | Corregimiento Caimalito, Pereira, Risaralda | 58.2 |
| Preserved Guadua | Vereda El Laurel, Quimbaya, Quindío—Dosquebradas, Risaralda | 54.5 |
| Clay Brick | Corregimiento Cerritos, Pereira | 23.1 |

2.2.2. Life Cycle Inventory (LCI)

Considering the phases and stages of the life cycle established in the system boundaries for the process inventory analysis of the CSB house, the calculation of material quantities was performed, as summarised in Table 2. An inventory structure comparable to that of the GaKB house was modelled in openLCA, and the inputs of both models were fed with the corresponding values.

**Table 2.** Life Cycle Inventory (LCI) of CSB and GaKB houses.

| Materials | CSB House (41.04 m$^2$) kg/house | GaKB House (60.65 m$^2$) kg/house |
|---|---|---|
| Round GaK culms | - | 1772.64 |
| GaK mats | - | 2048.48 |
| Concrete | 28,057.89 | 17,651.4 |
| Clay brick | 12,174.1 | 564.48 |
| Cement mortar | 1944.19 | 5189.6 |
| Steel * | 1047.15 | 515.51 |

* Steel includes rebar and roof structure.

*2.3. Impact Assessment*

Methodologically speaking, this phase implies five steps: (1) software feed; (2) environmental indicator's calculation (carbon footprint); (3) assessment results; (4) normalisation to functional unit; and (5) systematisation of results. In order to provide in-depth insight and clarity, the results of these items will be addressed in Section 3.

*2.4. Interpretation*

Interpretation as a process occurs permanently throughout the LCA; even the final phase involves four steps: (1) process impacts analysis; (2) comparisons analysis; (3) verification; and (4) reporting and communication. As with the previous subsection, these items will be addressed in Section 3.

**3. Results and Discussion**

This section showcases the results of the third and fourth phases of the LCA, corresponding to the impact assessment and interpretation processes.

*3.1. Environmental Indicator Calculation: Carbon Footprint*

The LCA results of the construction of the CSB housing typology show that it generates total emissions of 12,248 kg $CO_2$-eq. The LCA of the GaKB housing resulted in total emissions of 7220.3 kg $CO_2$-eq. Figure 7 shows the breakdown of the impact by process in each case.

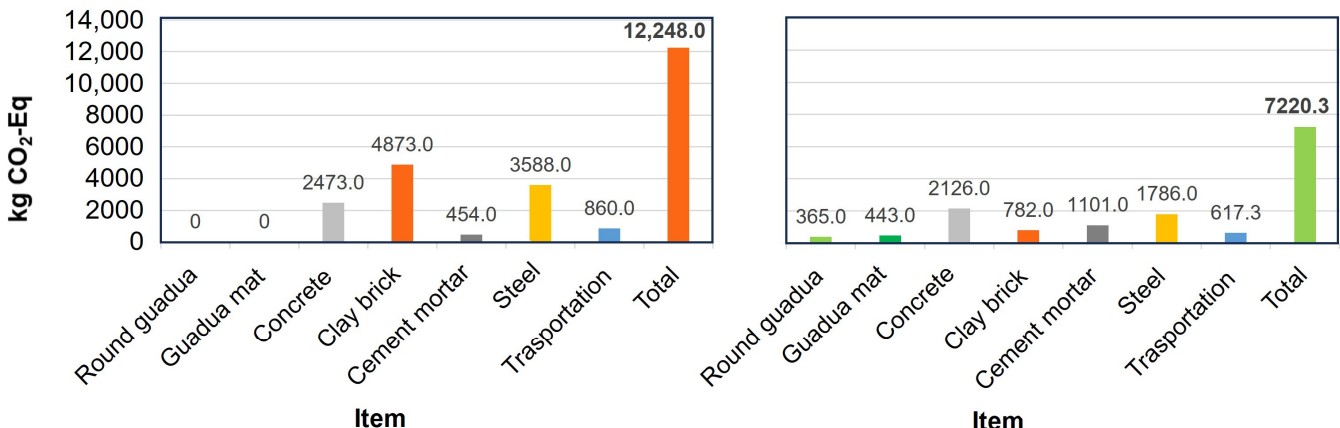

**Figure 7.** Emissions kg CO2-eq: (**left**) CSB house A = 41.04 m$^2$; (**right**) GaKB house A = 60.65 m$^2$.

*3.2. Normalisation to Functional Unit*

To perform the comparative analysis of the carbon footprint of the two housing typologies, the LCA results were normalised to the functional unit per living area, kg $CO_2$-eq/m$^2$. Considering that the CSB house has an area of 41.04 m$^2$, determining its environmental impact in terms of the functional unit shows that the carbon footprint for 1 m$^2$ of living area of the CSB house is 298.4 kg $CO_2$-eq/m$^2$. The GaKB house has a larger area than the CSB house, corresponding to 60.65 m$^2$; consequently, when determining its environmental impact in terms of the functional unit, the resulting carbon footprint per unit of housing area is 119.0 kg $CO_2$-eq/m$^2$. It is evident that the GaKB house has a significantly lower environmental impact since it accounts for about 40% of the carbon footprint of the CSB (Table 3).

**Table 3.** Comparison of material and kg $CO_2$-eq/m$^2$ of house type.

| Materials | CSB House (41.04 m$^2$) | | | GaKB House (60.65 m$^2$) | | | GaKB/CSB Percentage Comparison (%) |
|---|---|---|---|---|---|---|---|
| | kg/house | kg $CO_2$-eq | kg $CO_2$-eq/m$^2$ | kg/house | kg $CO_2$-eq | kg $CO_2$-eq/m$^2$ | |
| Round GaK culms | 0.0 | 0.0 | 0.0 | 1772.6 | 365.0 | 6.0 | - |
| GaK mats | 0.0 | 0.0 | 0.0 | 2048.5 | 443.0 | 7.3 | - |
| Concrete | 28,057.9 | 2473.0 | 60.3 | 17,651.4 | 2126.0 | 35.1 | 58.2 |
| Clay brick | 12,174.1 | 4873.0 | 118.7 | 564.5 | 782.0 | 12.9 | 10.9 |
| Cement mortar | 1944.2 | 454.0 | 11.1 | 5189.6 | 1101.0 | 18.2 | 164.1 |
| Steel * | 1047.2 | 3588.0 | 87.4 | 515.5 | 1786.0 | 29.4 | 33.7 |
| Transportation | | 860.0 | 21.0 | | 617.3 | 10.2 | 48.6 |
| Total | 43,223.3 | 12,248.0 | 298.4 | 27,742.1 | 7220.3 | 119.0 | 39.9 |

\* Steel includes rebar and roof structure.

*3.3. Comparative Carbon Footprint of CSB House vs. GaKB House*

The comparative graph of kg $CO_2$-eq by type of housing (Figure 8) shows that in both cases, the highest impacts on the environment are linked to technical materials (man-made); in the case of the CSB house, the element that generates the highest environmental load is precisely the structural brick, while in GaKB housing, the component that most affects the environment is the concrete used in the foundation and ground floor slab, due to the environmental loads from the production and transportation process of cement. By contrast, the biobased materials used in GaKB house (i.e., GaK) in the form of round Guadua and Guadua mats, which are the main components of the structural cemented bahareque system, are the ones that generate the least environmental load.

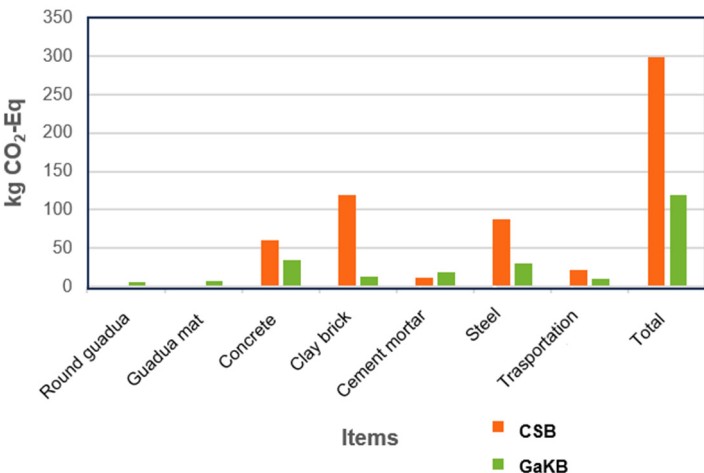

**Figure 8.** Carbon footprint comparison by house type.

*3.4. Process Impact Analysis*

In the CSB house, the highest impact corresponds precisely to the component that gives it its name as a construction typology, the clay brick (118.7 kg $CO_2$-eq/$m^2$), which is almost equal to that of the total GaKB house (119.0 kg $CO_2$-eq/$m^2$) (Table 3). Overall, two and a half houses built using GaK would have the same footprint as one CSB house. The high impact of the CSB house is largely due to energy consumption and the generation of greenhouse gases (GHGs) during the moulding and firing processes of the bricks [31,32].

The second process that generates the most global warming impact in the CSB house is associated with steel, which generates 87.4 kg $CO_2$-eq/$m^2$, whilst in the GaKB house, the use of steel generates only 29.4 kg $CO_2$-eq/$m^2$ (Table 3), corresponding to one third of the total impact of the GaKB house.

The third process in terms of global warming impacts relates to the use of concrete, which in the CSB house is 60.3 kg $CO_2$-eq/$m^2$, while in the GaKB house, it is 35.1 kg $CO_2$-eq/$m^2$, which constitutes 58.2% of the impact generated by the GaKB house (Table 3).

The significant reduction in impacts is due to the efficiencies allowed by the light cement bamboo frame (LCBF) system. For instance, the mass of steel reinforcement and concrete for the foundation and the ground floor slab of the GaKB house is lower than in the CSB house. The total mass of concrete for foundations and slabs per $m^2$ in the GaKB house accounts for 43% of the CSB house, whilst the mass of steel for reinforcement and roof in the CSB house is three-fold the mass of steel in the GaKB house (33%). Also, in the LCBF system, the steel purlins or profiles supporting the roof are replaced by round Guadua beams, which have much less impact.

The fourth process relating to global warming impacts in the CSB house is transportation, which accounts for almost double (21.0 kg $CO_2$-eq/$m^2$) the impact in the GaKB house (10.2 kg $CO_2$-eq/$m^2$), as seen in Table 3. Transportation constitutes an item sensitive to the availability of materials near the construction site [33]. In this case, the fact that both bricks and Guadua are produced locally and regionally favourably impacted the low carbon footprint and substantiate a principle of vernacular architecture in terms of the appeal of using locally produced materials [2,3].

A process that creates a high global warming impact in the GaKB house includes the cement mortar (18.2 kg $CO_2$-eq/$m^2$), which is used for rendering the walls and to fill in the internodes of Guadua poles at joining points. By contrast, ranking at the bottom of the global warming potential are the Guadua mats (7.3 kg $CO_2$-eq/$m^2$) and the round Guadua culms (6.0 kg $CO_2$-eq/$m^2$) (Table 3), which are exclusive to the GaKB house.

As seen in Table 4, leaving the clay brick aside, the manufacturing and construction processes associated with materials of high industrial transformation, such as concrete, cement mortar, and steel, together generate high impacts in both housing typologies. In the CSB house, the impact is 158.8 kg $CO_2$-eq/$m^2$, corresponding to 53.2% of the carbon

footprint, and in the GaKB house, it is 82.7 kg $CO_2$-eq/m$^2$, which is 69.5% of the carbon footprint before applying the $CO_2$ balance. This constitutes valuable technical and scientific evidence in favour of the replacement of energy- and carbon-intensive technical materials such as concrete, steel, and bricks by locally grown renewable materials in housing projects.

**Table 4.** Analysis of technical materials used in both houses.

| Materials | CSB House (41.04 m$^2$) | | GaKB House (60.65 m$^2$) | |
|---|---|---|---|---|
| | kg CO$_2$-eq/m$^2$ | Percentage (%) | kg CO$_2$-eq/m$^2$ | Percentage (%) |
| Concrete | 60.3 | 20.2 | 35.1 | 29.5 |
| Cement mortar | 11.1 | 3.7 | 18.2 | 15.3 |
| Steel * | 87.4 | 29.3 | 29.4 | 24.7 |
| Sub-total | 158.8 | 53.2 | 82.7 | 69.5 |
| Total | 298.4 | 100 | 82.8 | 100 |

* Steel includes rebar and roof structure.

It is also important to note that the mass of 3821.1 kg corresponding to the structural support materials in the GaKB house, represented by the round Guadua and the Guadua mat, is about 70% lower than the mass of the clay brick used in the CSB house, which is 12,174.1 kg, although the latter has a smaller area (Table 5). The mass of the clay brick of the CSB house is 296.64 kg/m$^2$ of living space, while that of the biobased round Guadua and Guadua mat of the GaKB house is equivalent to 63.0 kg/m$^2$ of living space (21.2%), evidencing its structural efficiency.

**Table 5.** Base materials for the structural system.

| Materials | CSB House (41.04 m$^2$) | | GaKB House (60.65 m$^2$) | | Percentage (%) |
|---|---|---|---|---|---|
| | kg/house | kg/m$^2$ house | kg/house | kg/m$^2$ house | |
| Round GaK culm | - | - | 1772.6 | 29.23 | - |
| GaK mats | - | - | 2048.5 | 33.77 | - |
| Clay brick | 12,174.1 | 296.64 | - | - | - |
| Total | 12,174.1 | 296.64 | 3821.1 | 63.00 | 21.2 |

### 3.5. Carbon Footprint Balance

To carry out the $CO_2$ balance by house typology, first the calculation of kg $CO_2$ captured in the GaKB house was carried out due to its intensive use of GaK, which during its growth process in the forest captures carbon as follows:

- A total of 126.41 t/ha of $CO_2$ are stored per hectare;
- Each hectare has 4050 Guadua culms;
- The mass of a Guadua plant (leaves, branches, root, and culm) is 31.21 kg on average [29,30].

Furthermore, a Guadua culm represents 67% of the Guadua's plant biomass [34], and it is also estimated that from each Guadua in the forest, an average of 12 m (lineal metres) of round Guadua or Guadua mats can be used for construction (Table 6).

**Table 6.** Calculation of kg $CO_2$ captured in the GaKB house.

| Materials | Guadua Poles (m) * | Bamboo Demand | | Guadua Poles/ m$^2$ | kg CO$_2$ Captured | | | kg CO$_2$/m$^2$ house |
|---|---|---|---|---|---|---|---|---|
| | | Guadua Lengths (m) | Guadua Poles (No.) | | kg CO$_2$/Guadua | % Culm | kg CO$_2$/house | |
| Round GaK | 295.44 | 12.00 | 24.62 | 0.41 | −31.21 | 67 | −514.82 | −8.49 |
| GaK mat | 118.00 | 12.00 | 9.83 | 0.16 | −31.21 | 67 | −306.90 | −3.39 |
| Total bamboo | 413.44 | 12.00 | 34.45 | 0.57 | −31.21 | 67 | −1075.29 | −11.88 |

* Lineal metres of Guadua poles.

The balance shows that thanks to the $CO_2$ captured in the growth process of the GaK, the carbon footprint of the GaKB house decreases by 10%, equivalent to 11.88 $CO_2$/m$^2$

(Table 7), which leads to the final comparison of the carbon footprint (Figure 9). Therefore the footprint of the GaKB house is further decreased from 39.9% to 35.9% of that of the CSB house (Table 7).

**Table 7.** $CO_2$ balance by type of house.

| Housing Typology | kg $CO_2$-eq/m$^2$ | GaKB/CSB Comparison (%) | kg $CO_2$/m$^2$ Captured | Balance kg $CO_2$-eq/m$^2$ | GaKB/CSB Comparison (%) |
|---|---|---|---|---|---|
| CSB | 298.44 | 100 | 0 | 298.44 | 100 |
| GaKB | 119.05 | 39.9 | −11.88 | 107.17 | 35.9 |

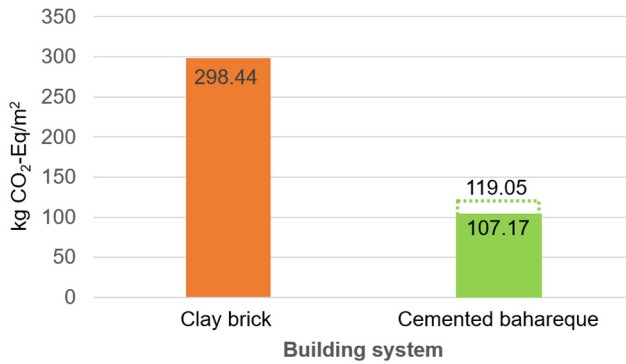

**Figure 9.** Comparative carbon footprint considering $CO_2$ balance.

## 4. Conclusions and Future Research

As demonstrated by this LCA study, bamboo-based building systems, such as the Lightweight Cement Bamboo Frame (LCBF), have a great potential to contribute to the reduction of the carbon footprint of housing construction in countries of the Global South with sizable bamboo and biobased resources, such as Colombia. An LCBF house using Guadua bamboo in Colombia has a carbon footprint that accounts for about 36% of the carbon footprint of a house using conventional materials such as bricks, concrete, and steel (CSB).

Bamboo construction inspired by vernacular techniques like the LCBF system requires less steel and concrete for foundations, ground floor slabs, and roofs than in conventional construction systems. Concrete foundations and slabs in LCBF systems are 43% lighter than in houses such as the CSB, using conventional materials such as bricks, concrete, and steel, which contribute to about 16% of global carbon dioxide ($CO_2$) emissions. The mass of steel for reinforcement and roof in a building like the CSB house is three-fold the mass of steel in the GaKB house (33%). Thick cement mortar renders contribute greatly to the wall mass, and together with the foundations, they result in the highest negative environmental impact in LCBF systems. Murphy and colleagues [35] also highlight that the overall use of aggregates, cement, and steel contributes to about 95% of the environmental impact of Guadua bamboo construction systems. Hence, the replacement of cement with alternative binders in the system is a key point for environmental improvement. Lime is a widely available material that requires less energy-intensive production processes, offers improved breathability within the building, and behaves more elastically than cement [36]. However, more research on the use of lime as an alternative render material in LCBF systems is needed.

LCA results for the built GaKB house demonstrate that vernacular housing projects that preserve cultural heritage can also be resilient and climate-neutral. This paper sets a precedent for the establishment of targeted government policies and industry practices that preserve the cultural heritage and vernacular technologies in the CCLC region and in other emergent economies worldwide, whilst promoting future-proof and net-zero carbon construction. The implementation of government policies that promote the use of

vernacular building technologies can both preserve cultural heritage and create resilient, climate-neutral, and sustainable vernacular housing projects [37]. This would be possible with an appropriate policy framework, but also with government incentives, sustainable production and management practices of bamboo plantations, and associated processes. This can contribute to the regenerative development of the CCLC and other regions where they could be used, with lasting favourable effects on local economies and communities. In this sense, the recent approval of the so-called "Guadua Law" in Colombia in 2022 is extremely relevant. Strong trade organisations such as Fedeguadua can incentivise the commercial and sustainable use of Guadua and bamboos in bamboo-growing regions such as the CCLC. The Guadua law mandates that at least 30% of the new constructions for rural housing that are part of government programmes and that are carried out within the territory that makes up the CCLC must be in bamboo [38].

LCA studies of vernacular housing projects, such as this one, demonstrating the low environmental impact of this type of construction, provide firm technical and scientific evidence in favour of using locally grown renewable materials in housing projects in Colombia and elsewhere.

*Future Research*

Among future research directions, three fields of opportunity are identified: (1) LCA of bamboo and wood products and components; (2) bamboo structural engineering [39]; and (3) materials chemistry.

Firstly, it is highlighted that LCA studies of bamboo-based buildings such as this one favour models that consider the quantities of main building materials required for load-bearing and non-load-bearing elements under the assumption that finishing and furnishing elements are optional to be used by the builder or the user and are usually the same in all typologies. Evolving user preferences for eco- and/or biobased materials and the momentum for the Circular Economy approach create a favourable environment to develop new comparative LCA studies involving structural bamboo and wood-based products and components such as flooring, doors, and windows. This will enable assessment of their impacts on the environment and meaningful comparisons with other biobased materials and with their equivalent technical materials, including aluminium, steel, or plastics widely used in the construction industry, such as polyvinyl chloride (PVC) and polystyrene (PS).

In addition, with regard to the opportunities in the field of bamboo structural engineering for housing, there is the possibility of carrying out Research and Development and Innovation (R&D&I) projects implementing dual modular structural systems that efficiently combine the use of vernacular bamboo techniques and wood in the wall frames with boards of various cladding materials such as OSB, gypsum, and Tetra Pak, seeking to reduce cement use, optimise construction times, and increase user acceptance. Furthermore, considering the elements most exposed to humidity that current standards prescribe to be in reinforced concrete, the feasibility of the structural use of foundation beams and plinths prefabricated with recycled plastic can be explored and validated.

Finally, the chemistry of materials is an important area of research, insofar as it allows the exploration of alternatives to Portland cement mortar, both for the filling of the cores and for the rendering of bahareque walls; thus, lime and biobased composites may be able to fulfil the physical–mechanical properties required to advance in the substitution [40].

**Author Contributions:** Conceptualisation and methodology, C.E.R., J.A.M. and H.F.A.; software, C.E.R.; validation, J.A.M. and H.F.A.; formal analysis, C.E.R., J.A.M. and H.F.A.; investigation, C.E.R. and J.A.M.; resources, C.E.R., J.A.M. and H.F.A.; data curation, J.A.M. and H.F.A.; writing—original draft preparation, C.E.R.; writing—review and editing, C.E.R., J.A.M. and H.F.A.; visualisation, C.E.R.; supervision, J.A.M.; project administration, C.E.R. and J.A.M.; funding acquisition, C.E.R., J.A.M. and H.F.A. All authors have read and agreed to the published version of the manuscript.

**Funding:** This research was funded by Colombian Ministerio de Ciencia Tecnología e Innovación (MINCIENCIAS) through Call 647 of 2014 and the Vice Rector's Office for Research, Innovation and Extension at Universidad Tecnológica de Pereira through grant number E2-18-1.

**Institutional Review Board Statement:** Not applicable.

**Informed Consent Statement:** Not applicable.

**Data Availability Statement:** Data are contained within the article.

**Acknowledgments:** This article presents part of the research results of the Doctoral Thesis in Environmental Sciences of the Faculty of Environmental Sciences of the Universidad Tecnológica de Pereira entitled "Resilience and sustainability of Vernacular Architecture: The Case of the Regional Architecture of Guadua Bahareque in the Coffee Cultural Landscape of Colombia". The authors would like to thank EDUR (Urban and Rural Development Company of Risaralda) for the provision of technical information on the building project 'Bello Horizonte'. We would also like to thank FEDEGUADUA, INNPULSA (the Colombian agency for entrepreneurship and innovation), Swisscontact, and SECO (the Swiss Economic Cooperation and Development Program) for the implementation of the project "Strengthening of the value chain of sustainable construction with Guadua in the area of the Colombian Coffee Cultural Landscape (CCLC)" (code PC+C011-19), which provided valuable information for this research. Finally, the authors thank Edwin Zea Escamilla, Senior Assistant at the Chair for Sustainable Construction ETH Zürich and M.Sc. student Howard Hurtado for their contributions to the review of the LCA models.

**Conflicts of Interest:** The authors declare no conflict of interest. The funders had no role in the design of the study; in the collection, analyses, or interpretation of data; in the writing of the manuscript; or in the decision to publish the results.

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
