# Peer review of "Bamboo Construction Inspired by Vernacular Techniques for Reducing Carbon Footprint: A Life Cycle Assessment (LCA)"

_sustainability, doi:10.3390/su152416893_

Round 1

Reviewer 1 Report

Comments and Suggestions for Authors

Dear authors,

I have carefully reviewed your article and I appreciate the effort you have put into it. I have some recommendations and questions to help you further enhance the quality and impact of your work:

1. How did you determine the percentage of carbon footprint reduction (10%) due to the CO2 captured during the growth of Guadua? Please provide more details on this calculation.

2. You briefly mentioned the efficiencies allowed by the light cement bamboo frame (LCBF) system. Could you provide more information on these efficiencies and how they contributed to reducing the environmental impact?

3. To enhance the context and significance of your research, include a paragraph in the introduction section that references other studies conducted in Colombia related to methodologies and tools for improving the construction sector in general. These studies could encompass a variety of methodologies beyond LCA. In this regard, include the following studies:

* Qualitative Analysis of Lean Tools in the Construction Sector in Colombia. https://doi.org/10.24928/2019/0185

* Circular Economy in the Construction Sector: A Case Study of Santiago de Cali (Colombia). https://doi.org/10.3390/su14031923

By doing so, you can provide a broader context for your work and demonstrate the significance of your contribution to the field.

4. Have you considered the impact of user preferences for eco- and/or biobased materials on the overall environmental footprint of housing in the region? It could be an interesting point to explore further.

5. Discuss the potential for replicating your study in other regions with similar vernacular housing systems to highlight the broader applicability of your research.

6. Verify Figure 6 reference.

I believe that addressing the points mentioned above will further enhance the robustness and applicability of your research.

Author Response

Dear reviewer 1

We are grateful for your recommendations and questions. We have taken all of them into account and are providing a response point by point below:

  1. How did you determine the percentage of carbon footprint reduction(10%) due to the CO2 captured during the growth of Guadua? Please provide more details on this calculation.

Answer. We considered reliable data from recent publications on carbon sequestration in Guadua plantations, based on which we made calculations considering standard yields for building components including Guadua-bamboo culms (poles) and Guadua-bamboo mats. The information was expanded in the article; please refer to section 3.6. Carbon Footprint Balance

  1. You briefly mentioned the efficiencies allowed by the light cement bamboo frame (LCBF) system. Could you provide more information on these efficiencies and how they contributed to reducing the environmental impact?

Answer. We expanded the information across the paper and included new references when referring to the LCBF system; i.e. please refer to line numbers 257 to 264.

  1. To enhance the context and significance of your research, include a paragraph in the introduction section that references other studies conducted in Colombia related to methodologies and tools for improving the construction sector in general. These studies could encompass a variety of methodologies beyond LCA. In this regard, include the following studies:

 * Qualitative Analysis of Lean Tools in the Construction Sector in Colombia. https://doi.org/10.24928/2019/0185

 * Circular Economy in the Construction Sector: A Case Study of Santiago de Cali (Colombia). https://doi.org/10.3390/su14031923

By doing so, you can provide a broader context for your work and demonstrate the significance of your contribution to the field.

Answer. Great suggestions, we included a paragraph in the introduction section that references other studies conducted in Colombia and the Global South; please refer to line numbers 84 to 89.  We also mentioned the relevance of circular economy and LCA in the discussion and conclusions.

  1. Have you considered the impact of user preferences for eco- and/or biobased materials on the overall environmental footprint of housing in the region? It could be an interesting point to explore further.

Answer. Indeed, it would be an interesting point to explore further in a publication focussing on user-preferences; however, it is outside the scope of this LCA study. We had already included a paragraph in the subsection “Future research” (line 372) considering user preferences for eco- and/or biobased materials as another topic to explore.

  1. Discuss the potential for replicating your study in other regions with similar vernacular housing systems to highlight the broader applicability of your research.

 Answer. We included a paragraph in the “Conclusions and Future research” section  discussing the potential for replicability of our study in other regions with similar vernacular housing systems i.e. the Global South.

  1. Verify Figure 6 reference.

Answer. We corrected the reference.

Reviewer 2 Report

Comments and Suggestions for Authors

The paper reports the life cycle assessment of bamboo constructions for carbon footprint reduction.

The authors perform LCA on a clay brick house and a bamboo house for social use, in Colombia.

The paper shows a particularly innovative study in line with the current problems of reducing energy consumption and mitigating environmental impact.

The methodology of the paper is clear and well described. The findings are supported by data. Future research directions are also interesting. However, I recommend specifying the objectives of the paper more clearly from the introduction, together with the novelty of the research and its importance in the era of climate change.

The graphs and tables are clear.

Please check that all references are inserted correctly in the text.

Author Response

 Dear reviewer 2

We are grateful for your recommendations and questions. We have taken all of them into account and are providing a response point by point below:

  1. The methodology of the paper is clear and well described. The findings are supported by data. Future research directions are also interesting. However, I recommend specifying the objectives of the paper more clearly from the introduction, together with the novelty of the research and its importance in the era of climate change.

Answer. We expanded on this in the introduction, specifying more clearly the objectives of the work, along with the novelty of the research and its importance in the era of climate change. We also included new references. This follows up on the Materials and Methods section were we also included: Figure 3. Study’s methodology flowchart.

  1. The graphs and tables are clear. Please check that all references are inserted correctly in the text.

Answer. We checked and corrected the references.

Reviewer 3 Report

Comments and Suggestions for Authors

1. Abstract section.

1.1. The novelty of the study must be added;

2. Introduction section;

2.1. Novelty must be clear;
2.2. Please add the contributions of the study at the end of this section;

3. Materials and Methods section

3.1. Please, more details of the methodology used have to be added;

3.2. A flowchart could be added to explain the implementation of the methodology;

3.3. Is any computational platform used to make the study?

4. The result section is poor. Please add the discussion and explanation of it

5. The conclusion section must be separated from the discussion section.

Author Response

Dear reviewer 3

We are grateful for your recommendations and questions. We have taken all of them into account and are providing a response point by point below:

  1. Abstract section.

1.1. The novelty of the study must be added;

Answer. We added the novelty of the research and its importance in the era of climate change.

  1. Introduction section;

2.1. Novelty must be clear;

Answer. We expanded the introduction, specifying more clearly the novelty of the research (i.e. line 103 to 105) and its importance in the era of climate change. We also included new references.

2.2. Please add the contributions of the study at the end of this section;

Answer. We added the contributions of the study at the end of the Introduction section; please refer to line numbers 112-117.

  1. Materials and Methods section

3.1. Please, more details of the methodology used have to be added;

Answer. We expanded the explanation of the methodology specifying more clearly the objectives of the work, along with the novelty of the research and its importance in the era of climate change. We also included new references. We also included: Figure 3. Study’s methodology flowchart, which provide a better idea of the methodology (i.e. line number 127).

3.2. A flowchart could be added to explain the implementation of the methodology;

Answer. Great idea, thanks. We added a flowchart to explain the implementation of the methodology (i.e. line number 127).

3.3. Is any computational platform used to make the study?

Answer. Yes, we used the software openLCA v 1.10.3 and the ecoinvent 3.6 database to undertake the LCA study.

  1. The result section is poor. Please add the discussion and explanation of it

Answer. The results section has been integrated with the discussion section to provide thorough analysis of the results.

  1. The conclusion section must be separated from the discussion section.

Answer. We separated them.

Reviewer 4 Report

Comments and Suggestions for Authors

The present manuscript presents an interesting work in comparing the environmental impact of a conventional clay brick house (CBH) and Guadua angustifolia Kunth bahareque (GaKB) houses. However, using software that may be a "black box" leads to an unclear understanding of the procedure to determine the carbon footprint. Some conclusions follow common sense rather than the results obtained. I suggest improving and explaining the methodology section, such as the main parameters used and fed to the software UMBERTO and others used. After that, the manuscript may be used as a reference for readers in the area. 

Author Response

Dear reviewer 4,

We are grateful for your recommendations and questions. We have taken all of them into account and are providing a response point by point below:

The present manuscript presents an interesting work in comparing the environmental impact of a conventional clay brick house (CBH) and Guadua angustifolia Kunth bahareque (GaKB) houses. However, using software that may be a "black box" leads to an unclear understanding of the procedure to determine the carbon footprint. Some conclusions follow common sense rather than the results obtained. I suggest improving and explaining the methodology section, such as the main parameters used and fed to the software UMBERTO and others used. After that, the manuscript may be used as a reference for readers in the area.

Answer. We expanded the explanation of the methodology and added a flowchart to explain its implementation. We did not use the software ‘UMBERTO’, we used the software openLCA v 1.10.3 and the ecoinvent 3.6 database to undertake the study. The boundary conditions and rational of the study are explained in the Materials and Methods section, along with the parameters used and fed to the software.

Reviewer 5 Report

Comments and Suggestions for Authors

This is a very novel topic, the author uses the LCA method to compare the emissions of several different types of residential buildings, which is expected to support the widespread application of bamboo in emission-reduction buildings. However, there are still some objectivity problems in this manuscript, and it is suggested that the author should be patient and correct them in order to improve the academic level and depth of this manuscript.

Comments on the Quality of English Language

This is a very novel topic, the author uses the LCA method to compare the emissions of several different types of residential buildings, which is expected to support the widespread application of bamboo in emission-reduction buildings. However, there are still some objectivity problems in this manuscript, and it is suggested that the author should be patient and correct them in order to improve the academic level and depth of this manuscript.

Author Response

Dear reviewer 5

We are grateful for your recommendations and questions. We have taken all of them into account and are providing a response point by point below:

This is a very novel topic, the author uses the LCA method to compare the emissions of several different types of residential buildings, which is expected to support the widespread application of bamboo in emission-reduction buildings.

However, there are still some objectivity problems in this manuscript, and it is suggested that the author should be patient and correct them in order to improve the academic level and depth of this manuscript.

Answer. In the spirit of objectivity, we have taken into account your grades in the evaluation criteria as a reference to improve the article; please see below.

  1. Are all the cited references relevant to the research?

Answer. We believe so. In any case, we expanded on the information provided and included new references about the efficiencies allowed by the light cement bamboo frame (LCBF) system and how they contributed to reducing the environmental impact. Additionally, we included references about other studies conducted related to methodologies and tools for improving the construction sector in general and in Colombia and the Global South. There is also mention of sustainable construction as a response to climate change and how vernacular housing projects that preserve the cultural heritage can also be resilient and climate-neutral.

  1. Are the research design, questions, hypotheses and methods clearly stated?

Answer. We expanded ON the introduction, specifying more clearly the objectives of the work, along with the novelty of the research and its importance in the era of climate change.

We also expanded the explanation of the methodology specifying more clearly the objectives of the work and we included a flowchart to explain its implementation. As such, Figure 3. Study’s methodology flowchart provides a better idea of the systematic approach of the study (i.e. line number 127).

As we used the software openLCA v 1.10.3 and the ecoinvent 3.6 database to undertake the study, we explained parameters used and fed to the software.

  1. Are the arguments and discussion of findings coherent, balanced and compelling? &
  2. For empirical research, are the results clearly presented?

Answer. We reorganised and expanded the results and discussion section in order to gain clarity and objectivity. We also included a paragraph in the “Conclusions and Future research” section discussing the potential for replicability of our study in other regions with similar vernacular housing systems i.e. the Global South.

  1. Is the article adequately referenced?

Answer. We added a few references in the introduction and in the discussion and conclusions sections, and can confirm that all citations are recent, relevant and well referenced.

  1. Are the conclusions thoroughly supported by the results presented in the article or referenced in secondary literature?

Answer. We expanded the conclusions section rigorously supported on the results presented and we included a section on “Future research” directions.

Round 2

Reviewer 1 Report

Comments and Suggestions for Authors

The authors have addressed the suggestions provided I believe the paper can be accepted for publication in the current state 

Reviewer 3 Report

Comments and Suggestions for Authors

The authors made all the indications that i asked for, hence, the paper is ready to be published.

Reviewer 4 Report

Comments and Suggestions for Authors

The suggestions and questions have been attented, so the manuscript has been improved and is ready for publication.

Reviewer 5 Report

Comments and Suggestions for Authors

The author carefully modified the reviewer's comments on the first edition of the manuscript. After careful review, the reviewer found that the manuscript met the acceptance criteria of the journal and suggested direct acceptance.